# Health and Environmental Hazards of the Toxic *Pteridium aquilinum* (L.) Kuhn (Bracken Fern)

**DOI:** 10.3390/plants13010018

**Published:** 2023-12-20

**Authors:** Matěj Malík, Otakar Jiří Mika, Zdeňka Navrátilová, Uday Kumar Killi, Pavel Tlustoš, Jiří Patočka

**Affiliations:** 1Department of Agroenvironmental Chemistry and Plant Nutrition, Faculty of Agrobiology, Food and Natural Resources, Czech University of Life Sciences Prague, Kamýcká 129, 165 00 Praha 6-Suchdol, Czech Republic; malikmatej@af.czu.cz (M.M.); tlustos@af.czu.cz (P.T.); 2Department of Crisis Management, Faculty of Security Management, Police Academy of the Czech Republic, Lhotecká 559/7, 143 01 Praha 4, Czech Republic; 3Department of Radiology, Toxicology and Civil Protection, Faculty of Health and Social Studies, University of South Bohemia, J. Boreckého 1167/27, 370 11 České Budějovice, Czech Republic; udaykumarkilli9@gmail.com (U.K.K.); toxicology@toxicology.cz (J.P.); 4Department of Botany, Faculty of Science, Charles University, Benátská 433/2, 128 00 Praha 2, Czech Republic; zdenka.navratilova@natur.cuni.cz; 5Department of Chemistry, Faculty of Science, University of Hradec Králové, Hradecká 1285, 500 03 Hradec Králové, Czech Republic

**Keywords:** ptaquiloside, weed, bracken fern, fiddleheads, bioactive compounds, *Pteridium aquilinum*, thiaminase, health

## Abstract

Bracken fern (*Pteridium aquilinum* (L.) Kuhn) is ubiquitous and acts as a cosmopolitan weed in pastures and similar environments. Despite its historical uses, it presents risks due to toxicity. This study, conducted in the second half of 2023, aimed to assess the environmental and health hazards of *P*. *aquilinum*, primarily focusing on its carcinogenic compound, ptaquiloside. The literature was comprehensively reviewed using diverse databases, including PubMed, Web of Science, Scopus, and Google Scholar. Information was synthesized from original research articles, meta-analyses, systematic reviews, and relevant animal studies. Animals grazing on bracken fern face annual production losses due to toxin exposure. The substantial impact on biodiversity, animal health, and human well-being arises from the presence of ptaquiloside and related compounds in milk, meat, and water, along with the increasing global prevalence of *P*. *aquilinum* and its swift colonization in acidic soil and fire-damaged areas. The objectives were to identify major bioactive compounds and explore their effects at molecular, cellular, pathological, and population levels. Various cooking techniques were considered to mitigate toxin exposure, although complete elimination remains unattainable. Therefore, the findings emphasize the need for cautious consumption. In conclusion, continued research is necessary to better understand and manage its environmental and health implications.

## 1. Introduction

### 1.1. Health and Environmental Hazards

Bracken fern (*Pteridium aquilinum*) (Figure 1) has long been recognized as a poisonous plant posing a threat to free-ranging cattle. The secondary metabolites of this fern exhibit high biological activity, leading to various chronic and acute health issues in diverse animal species, including humans [1,2]. The International Agency for Research on Cancer classified bracken fern as possibly carcinogenic to humans (Group 2B) [3,4]. Its primary and most hazardous metabolite is ptaquiloside, with direct or indirect human intake occurring mainly through the milk of animals grazing on bracken fern [5,6].

### 1.2. Cultural Significance and Culinary Applications

Despite the toxicity present in all parts of the plant, the rhizome historically served as a starchy food source, especially during times of famine. The leaves were utilized for silage in livestock and for packaging fruit. Additionally, bracken fern found applications in roofing, brewing, glass making, and as cattle feed [5,7]. *P*. *aquilinum* has been consumed by various cultures throughout history, particularly in Asia, Africa, and parts of Europe. Advocates of its edibility emphasize its abundant availability, nutritional value, and cultural significance. In some regions, this fern is still cultivated as a food and is especially popular in Japan. Young, still-curled fern shoots and leaves, known as “fiddleheads”, are a culinary specialty used in traditional dishes, gaining popularity for their unique taste and texture [8,9,10]. They are consumed raw in salads or cooked and prepared in various ways, providing a potential dietary supplement rich in vitamins, minerals, and fiber. Some products, particularly those based on rhizomes, were found to be free of ptaquiloside. However, the content may vary by season, with elevated levels primarily observed in autumn. Rhizomes and rhizome-based products could be considered safe regarding ptaquiloside toxicity [1].

### 1.3. Cooking Techniques and Safety Considerations

While some argue for the safe consumption of bracken fern after proper preparation, others caution against its intake due to potential health risks. Advocates of consumption stress the importance of adequate preparation and cooking techniques to minimize potential health risks [11]. Ptaquiloside can be partially degraded, and certain toxic protein compounds, such as thiaminase enzymes, decompose during the cooking process at high temperatures [12].

Various cooking techniques can contribute to reducing ptaquiloside levels in *P*. *aquilinum*, potentially enhancing its safety for consumption. High-temperature methods such as grilling, frying, and roasting can aid in breaking down ptaquiloside, with their effectiveness dependent on the intensity and duration of heat exposure. Blanching and boiling the fern in water can also reduce ptaquiloside content by allowing these compounds to leach into the water [2]. On the other hand, steaming is an effective cooking method that softens the fern while preserving its nutritional value and reducing toxin content [13]. Simultaneous saccharification and fermentation, involving the use of snail digestive juice and yeast, represent another promising approach for detoxifying poisonous plants such as bracken fern, albeit requiring specific optimization of techniques and conditions [14]. Bracken starches have been observed to have a lower viscosity than most rice and potato varieties, as well as a lower gelatinization temperature than cereal starch. Compared to rice and potato starches, bracken starches could potentially form much softer and stickier gels, indicating their potential utilization in developing both food and nonfood products [15].

It is essential to note that no cooking method can eliminate ptaquiloside and other toxic agents from bracken fern. Therefore, it is advisable to limit its consumption, especially for vulnerable populations such as pregnant women and children [16,17,18].

### 1.4. Comprehensive Review Approach

Compared to other recent reviews on a similar topic [19,20,21,22,23], this review offers a distinct, broader, and more nuanced perspective based on current literature data. It comprehensively explored the botanical, chemical, pharmacological, and toxicological properties, along with veterinary and human medical aspects of the bracken fern. Our research included original research articles, meta-analyses, systematic reviews, and relevant animal studies. To ensure a comprehensive approach, we consulted various scientific databases, including PubMed, Web of Science, Scopus, and Google Scholar, covering fields such as botany, pharmacology, and toxicology. We employed diverse keywords related to the bracken fern, encompassing aspects such as “*Pteridium aquilinum*”, “bracken fern”, “Pteridophyta”, “cyanogenic compounds”, “ptaquiloside”, “toxicity”, “phytochemicals”, “pharmacological activity”, and “allelopathy”. This review integrated information across different periods, providing a holistic overview that spanned from historical uses of the bracken fern, incorporating older references, to the latest research findings. The literature search embraced a broad timeframe, encompassing historical sources through to more recent publications.

## 2. Botanical Aspects of the Plant

*P*. *aquilinum* is a globally distributed fern species [24,25,26]. The original genus name, *Pteris*, is derived from the Greek word *pterón*, meaning “feather” or “wing”, referencing the spreading blade of the fern’s fronds. Pteridium also originates from the Greek word *pterídios*, signifying “feathered”. The plant’s species name is linked to the Latin word *aquila*, translating to “eagle”. However, the origin of this species’ name is still debated. Generally believed to stem from the leaf shape resembling an eagle’s wing [27], surviving writings indicate that Carl Linnaeus associated it with a specific arrangement of vascular bundles in a root’s transverse section, suggesting a double-headed eagle, often seen in coats of arms [28]. Friedrich Adalbert Maximilian Kuhn assigned the current binomial name in 1879 [29].

### 2.1. Taxonomic Complexity: The Diverse Subspecies of P. aquilinum

Bracken fern or eagle fern belongs to the phylum Pteridophyta and the family Dennstaedtiaceae (formerly Polypodiaceae). The Dennstaedtiaceae family encompasses multiple genera and species, constituting a diverse group with various fern species sharing evolutionary ancestry [30].

Within the genus *Pteridium*, there are several subspecies of *P*. *aquilinum*. However, it is important to note that *P*. *aquilinum* is generally considered a single species with multiple subspecies, rather than separate species within the genus [31]. According to recent taxonomic classification, this species has 12 subspecies with several varieties and morphotypes [32]. These subspecies are:*Pteridium aquilinum* subsp. *Aquilinum*;*Pteridium aquilinum* subsp. *Wightianum*;*Pteridium aquilinum* subsp. *Capense*;*Pteridium aquilinum* subsp. *Centrali-africanum*;*Pteridium aquilinum* subsp. *Latiusculum*;*Pteridium aquilinum* subsp. *Occidentale*;*Pteridium aquilinum* subsp. *Pubescens*;*Pteridium aquilinum* subsp. *Aquiliniforme*;*Pteridium aquilinum* subsp. *Pteridium*;*Pteridium aquilinum* subsp. *Caudatum*;*Pteridium aquilinum* subsp. *Pseudocaudatum*;*Pteridium aquilinum* subsp. *Pinetorum*.

Bracken fern is the common name used for all these subspecies [22,33]. Fossil evidence suggests the plant is at least 55 million years old [34]. The subspecies’ systematics were based on the degree of hairs of rhizomes, roots, and leaves and their distribution on the plant, the angle of the midvein of individual leaf orders, or the inner and outer indusium size [35]. The complexity in subspecies classification reflects the diverse morphological features exhibited by bracken fern across its global distribution, making it a challenging taxonomic group to study and categorize [30].

### 2.2. Morphological Features and Global Distribution

*P. aquilinum* possesses robust hairy rhizomes, serving as storage organs for accumulating carbohydrates, reaching diameters of up to 1.2 cm. From these rhizomes, slender black roots extend more than 50 cm into the soil [36]. The branched rhizome produces deciduous leaves, characterized by light green triangular structures measuring 1–2.5 m in length. The blade is composed of 2 to 4 pinnate leaflets, featuring large leaflets with an arched structure and curled leaf edges [37]. Clusters of sporangia, known as sori, are situated on the leaflet undersides, protected by indusium and modified leaf margins. Spores, appearing between July and August in temperate zones, are yellowish-brown, very fine, and tetrahedra-spherical [38,39].

The bracken fern exhibits a widespread global distribution, recognized as an expansive species due to its ability to thrive across diverse environments [40]. While not invasive in the traditional sense, bracken fern can become problematic in specific regions where it forms dense stands, outcompeting other plant species, especially in areas with nutrient-poor, acidic soils, and increased humidity [25]. It is most prevalent in temperate and subtropical regions worldwide and found on all continents except Antarctica, showcasing its adaptability to various environmental conditions [41]. Its growth spans all forest zone countries, reaching up to the upper forest border. Thriving in nutrient-poor, acidic soils with heightened humidity, the fern often forms extensive continuous stands, penetrating the soil with numerous thick creeping rhizomes that extend into the surroundings [42]. Ferns, particularly *P. aquilinum*, have successfully colonized diverse regions by playing a crucial role in modifying soil ecosystems, particularly in nitrogen acquisition. Their ability to meet nitrogen requirements independently of soil nitrogen levels provides them a competitive advantage over native plant species [43]. Previous research has demonstrated that ferns such as *P. aquilinum* can stimulate and influence nitrogen turnover, resulting in higher concentrations of mineralizable nitrogen and NH_4_^+^ compared to other plants. The study has revealed an “open N cycle” in bracken fern soils, with elevated net nitrification and NO_3_^−^ accumulation. The soil under *P. aquilinum* experiences faster NH_4_^+^ to NO_3_^−^ conversion, leading to enhanced nitrogen mineralization and nitrification compared to neighboring vegetation [24,44,45].

### 2.3. Ecological Significance and Mycorrhizal Associations

Bracken fern exhibits an exceptionally high reproductive rate, robust regrowth capacity, and rapid adaptability to changing ecological conditions [46,47]. In certain regions, the fern is classified as a nuisance or weed due to its invasion of fields and pastures, facilitated by its resilient, partly low-lying rhizome that withstands burning and plowing [48]. The intricate associations between *P. aquilinum* and fungi exemplify a compelling instance of ecological symbiosis. Bracken fern engages in mutualistic mycorrhizal relationships with both arbuscular mycorrhizal and ectomycorrhizal fungi, offering nutrients, pathogen protection, and support for seedling growth and establishment [49,50,51].

Moreover, allelopathic interactions between bracken fern and fungi play a role in shaping soil microbial communities and influencing plant competition dynamics [52,53]. *P. aquilinum* impedes the establishment of pioneer species while favoring the germination of late-successional species. This ecological filtering is determined by shade and soil-mediated processes, contributing to arrested succession, as most seed inputs are from pioneers [54,55,56]. Fungal endophytes residing within bracken fern tissues produce bioactive compounds, including alkaloids, with ecological and medicinal significance. Understanding these interactions holds conservation implications, underscoring the importance of preserving these relationships for biodiversity and ecosystem resilience in the face of anthropogenic pressures such as habitat fragmentation and climate change [57].

## 3. Chemical Composition

Bracken fern boasts an intricate chemical profile that has captivated the interest of researchers and scientists. It serves as a rich repository of phytochemicals, encompassing phenolic compounds, flavonoids, alkaloids, tannins, and terpenoids [58]. These compounds play a pivotal role in the plant’s defense against pathogens and predators [59,60]. Phenolic compounds present in bracken fern, such as the flavonoids quercetin and kaempferol, showcase antioxidant properties, suggesting potential health benefits [48,61,62]. *P*. *aquilinum* encompasses various nutritional components, including carbohydrates, proteins, dietary fiber, vitamins, and minerals [63]. Although the nutritional composition may fluctuate based on factors such as plant maturity and environmental conditions, bracken ferns typically harbor significant quantities of dietary fiber, vitamin C, vitamin A, vitamin B complex, potassium, and manganese [64]. Additionally, various poisonous and bioactive compounds, such as cyanogenic glycosides, antithiamine factors, and compounds with carcinogenic activity, have been identified (Table 1) [18,46].

The principal cyanogenic compound in *P*. *aquilinum* is prunasin (Figure 2). It represents a cyanogenic glycoside comprising a sugar moiety (glucose) attached to a benzaldehyde (mandelonitrile), akin to amygdalin, with the molecular formula C_14_H_17_NO_6_, molar mass 295.29 g/mol, and a melting point of 147 to 148 °C [65]. For perspective, Venezuelan bracken croziers contain prunasin at concentrations ranging from 10.4 to 61.3 mg/g of fresh plant tissue [66], with the highest levels found in young fronds (Table 1). The potential release of up to 5.9 µg of hydrocyanic acid (HCN) from 1 g of fresh Venezuelan bracken underscores the low levels of prunasin found [18,67]. One gram of prunasin has the potential to release up to 96 µg of HCN [66]. To put these findings into context, the Committee on Toxicity of Chemicals in Food, Consumer Products, and the Environment (COT) notes that the World Health Organization and the Council of Europe have established tolerable daily intakes for cyanide at 12 and 20 µg/kg body weight, respectively. Additionally, COT has identified an acute reference dose (ARfD) of 5 µg/kg body weight, calculated by applying a 100-fold uncertainty factor to the lowest reliably observed acute lethal dose in humans, which is 0.5 mg/kg body weight. For a person weighing 60 kg, this would necessitate consuming more than 50 g of Venezuelan bracken to exceed the ARfD [68]. Furthermore, studies have demonstrated the presence of two groups of antithiamine compounds, namely thiaminase enzymes and thermostable compounds of antithiamine character. The identified compounds of the second-mentioned group include astragalin, caffeic acid, and isoquercetin [18,46,69].

In 1983, two groups of scientists from Japan and the Netherlands isolated a compound with mutagenic and carcinogenic effects from bracken fern [70,71]. The compound, named ptaquiloside, possesses a norsesquiterpene backbone with glycosidic moieties. It is a colorless, amorphous compound with the molecular formula of C_20_H_30_O_8_, a molar mass of 398.45 g/mol, and a melting point of 85 to 89 °C. Ptaquiloside demonstrates high solubility in water and fair solubility in organic solvents such as ethyl acetate [72]. Its content in bracken fern varies widely, commonly ranging from virtually zero to ∼13 mg/g of plant dry matter [20]. The concentration of ptaquiloside is subject to variations influenced by botanical factors, primarily the type and age of the plant organs [73,74]. The maximum ptaquiloside content in bracken fern occurs at emergence, with recorded contents reaching up to 37 mg/g in plant tissue [75]. Additionally, *P*. *aquilinum* contains other sesquiterpene *β*-glucosides of illudane-type, namely isoptaquiloside and caudatoside (Figure 3), which exhibit structural similarities with ptaquiloside [76,77]. Furthermore, various steroid compounds, specifically *α*- and *β*-ecdysones, have been identified [78].

**Table 1 plants-13-00018-t001:** Occurrence and range of bioactive compounds in different parts of bracken fern (*Pteridium aquilinum*).

Phytochemical Group	Main Active Compounds	Occurrence in Plant Parts	Range of Values	References
Illudane glycosides	Ptaquiloside	All parts	0–<13 (up to 37) mg/g	[1,20,75]
	Isoptaquiloside	Fronds	unquantified	[76,77]
	Caudatoside	Fronds	0–0.55 mg/g	[76,77]
Cyanogenic glycosides	Prunasin	Fronds, Rhizomes, Rachis	10.4–61.3 mg/g	[66,79]
Flavonoids	Quercetin	Fronds, Rhizomes	0.57–0.86 mg/g	[48,62,80,81]
	Kaempferol	Fronds, Rhizomes	1.1–2.55 mg/g	[18,48,62,82]
Antithiamine compounds	Thiaminases types 1 and 2	Fronds, Rhizomes	3.1 and 3.5 µg thiamine destroyed/g plant material/hour	[18,46,83]
	Astragalin	Fronds, Rhizomes	0–0.21 µg/g	[69,81,84]
	Caffeic acid	Fronds	unquantified	[69,85]
	Isoquercetin	Fronds	unquantified	[69,81]
Steroid compounds	*α*- and *β*-ecdysones	Fronds	unquantified	[78,86]

## 4. Pharmacological Activity and Toxicity

Bracken fern (*Pteridium aquilinum*) has a long history of traditional use in medicinal folklore [87]. Researchers have identified numerous bioactive compounds in bracken fern that demonstrate potential pharmacological activities. Some studies suggest that specific compounds in bracken fern, including flavonoids, alkaloids, and phenolic compounds, exhibit antimicrobial [88], antibacterial [89,90,91], antioxidative [64,91,92,93], immunomodulatory [92,94,95], and anti-inflammatory [93] properties [96,97].

### 4.1. Antimicrobial Potency: In Vitro Studies

Phenolic acids, known for their antimicrobial properties even at concentrations as low as 100 to 150 ppm *w*/*v* [98,99], are present in bracken fern sporophytes (0.25–0.75%, phenolic acids/g dry weight) and gametophytes (0.15–0.20%, phenolic acids/g dry weight) at sufficiently high levels to act as significant antimicrobial compounds. The concentrations of phenolic acids in *P. aquilinum* leaves vary seasonally, influenced by plant age and various environmental and biological factors. These phenolic acids differentially impede the growth of fern pathogens and other bacteria in vitro, persisting in plant tissues throughout most of the growing season and effectively inhibiting the proliferation of pathogenic microorganisms [88].

The minimum inhibitory concentration (MIC) for the *n*-hexane extract of leaves, determined via the tube dilution method, demonstrated inhibition at 300 to 350 mg/mL for *Staphylococcus aureus*, *Enterobacter aerogenes*, and *Pseudomonas aeruginosa*, and at 350 mg/mL for *Escherichia coli*. In contrast, there was no inhibition of the ethanolic extract for any of the organisms. The minimum bactericidal concentration (MBC) of the n-hexane extract for *P. aeruginosa*, *E. coli*, and *E. aerogenes* indicated susceptibility at 300 to 350 mg/mL [89].

The MIC test of the ethanolic and petroleum ether extract of *P. aquilinum* against bacterial pathogens (*Bacillus subtilis* and *S. aureus*) was observed as 1 mg/mL. For E. coli and *Proteus vulgaris*, it was found to be 0.8 mg/mL. This study revealed that extracts of bracken fern were either more effective or equally effective against the tested organisms except *P. vulgaris* compared to streptomycin [91]. The MIC values further showed that *P. aquilinum* essential oil contained 32.86% oxygenated monoterpenes and exhibited potent activity against *Erwinia amylovora* (0.625 μL/mL), followed by *Pectobacterium carotovorum* subsp. *carotovorum* (2.50 μL/mL) and *Pseudomonas savastanoi* pv. *savastanoi* (5.00 μL/mL) [90]. Further studies may lead to their use as safe alternatives to synthetic antimicrobial drugs.

### 4.2. Antioxidant Properties: In Vitro Insights and Prospects for In Vivo Exploration

The antioxidant activity of *P. aquilinum* leaves was slightly lower when compared with ascorbic acid (% inhibition in mg/mL). The scavenging activities of 2,2-diphenyl-1-picrylhydrazyl (DPPH) and 2,2′-azino-bis(3-ethylbenzothiazoline-6-sulfonic acid) (ABTS) radicals were 84 and 73.3%, respectively, for *P. aquilinum* leaves, while for ascorbic acid, it was 88.2% and 83%, respectively [91]. In a comparison of extracts from selected ferns (*Onoclea struthiopteris*, *Onoclea orientalis*, *Osmunda japonica*, and *P. aquilinum*), all fern extracts exhibited antioxidant effects in vitro. Notably, the roots of *O. japonica* and the fronds of *O. orientalis* were the most efficient [93].

Utilizing a ferric-reducing antioxidant power assay (FRAP), it was demonstrated that the purified polysaccharide obtained from *P. aquilinum* possessed strong reductive power (FRAP value: 827.6 μmol/L), comparable to that of vitamin C. Additionally, it exhibited moderate scavenging activities against DPPH radicals (83.1%) at an 800 μg/mL dose. These results suggest that the purified polysaccharide had a noticeable effect on scavenging free radicals, particularly at higher concentrations. However, its activity was not more potent than vitamin C at the same dose. The percentage inhibition of superoxide generation via 800 μg/mL doses of the crude polysaccharides, the purified polysaccharide, and vitamin C was found to be 42.2%, 60.5%, and 84.2%, respectively. The scavenging effects of the crude polysaccharides, the purified polysaccharide, and vitamin C on the self-oxidation of 1,2,3-phentriol concentration were dependently increasing. At 100 μg/mL doses, they were 23.3%, 52.4%, and 90.7%, respectively [64].

Similar results were also obtained by Zhao et al. [92] in their study when another new polysaccharide (named PAP-3) was derived from *P. aquilinum*. PAP-3 exhibited strong free-radical scavenging activity on DPPH and ABTS radicals in vitro, albeit slightly lower than that of vitamin C. These studies collectively demonstrated the robust antioxidant capacity of bracken fern compounds in in vitro antioxidation tests. Future research endeavors should focus on investigating the antioxidant activity of this plant through in vivo experiments.

### 4.3. Immunomodulatory and Anticancer Potential: Insights into Cellular Responses and Therapeutic Prospects

Immunostimulation is considered a crucial defense mechanism against various diseases. The immunomodulatory effect of PAP-3 on RAW264.7 cells, a macrophage cell line derived from a male mouse tumor induced via the Abelson murine leukemia virus, was examined across a concentration range of 0 to 200 μg/mL. PAP-3 demonstrated the ability to induce RAW264.7 cell proliferation at concentrations ranging from 12.5 to 200 μg/mL, with no detectable cytotoxic effects up to 200 μg/mL. Incubation with PAP-3 significantly increased nitric oxide (NO) production from RAW264.7 cells, reaching 17.31 μM at 25 μg/mL concentration, comparable to the positive control (lipopolysaccharide, 10.0 μg/mL) [92].

Immunomodulatory effects associated with prolonged bracken fern feeding and enzootic bovine hematuria (EBH) were assessed using various tests, including the tube agglutination test for *Brucella abortus* antibodies, dinitrochlorobenzene skin test, lymphocyte proliferation assay, histopathological examination of lymph nodes, and skin biopsies in cows. Cows subjected to prolonged bracken fern feeding and those affected by EBH exhibited a decrease in humoral and an increase in cell-mediated immune responses compared to controls [95].

Another study investigated the immunomodulatory effects of bracken fern through daily extract ingestion by murine hosts over 14 (or up to 30) days. In C57BL/6 mice administered the extract via gavage, histological analyses revealed a significant reduction in splenic white pulp area. Various immune response parameters/functions were assessed, demonstrating a reduction in delayed-type hypersensitivity and interferon-gamma production by natural killer cells during T helper 1 priming. The innate response, assessed via natural killer cell cytotoxic functionality, was also diminished. While bracken fern affected components of acquired and innate immune responses, not all responses were modified, as seen in the analyses of humoral response and macrophage activity. These results suggest the immunosuppressive effect of *P. aquilinum* in mice, potentially contributing to an increased risk of cancer formation in exposed hosts [94].

The study by Dion et al. [93] revealed decreased interleukin-1 beta (IL1-β) gene expression for *P. aquilinum* frond extracts. Indirect measurement of NO via inducible nitric oxide gene expression showed a 50% decrease with the extract of *O. orientalis* fronds at a low concentration (20 μg/mL) compared to *P. aquilinum* fronds (160 μg/mL) and leaves of *O. japonica*, with the latter showing a higher decrease at a high extract concentration (160 μg/mL).

Additionally, extracts from bracken fern have demonstrated the potential to inhibit the growth of specific cancer cells in in vitro studies [100,101]. Treatment of human cancer cell lines with bracken fern dichloromethane extracts induced DNA damage and apoptosis at high concentrations (200 μg/mL) and caused cell cycle arrest at milder concentrations (50 and 30 μg/mL), depending on the cell type and tumor origin [101]. Williams et al. [100] reported that ptaquiloside exhibited selective toxicity against cancer cells compared to noncancer retinal epithelial cells, indicating its potential as a therapeutic agent.

These findings underscore the potential of *P. aquilinum* as a source of natural compounds with therapeutic value. However, it is crucial to note that bracken fern also contains several toxic compounds and bleeding factors, especially thiaminases and ptaquiloside, found in high concentrations in the plant’s young parts (young fronds) and rhizomes [18,46].

### 4.4. Ptaquiloside: Carcinogenic Properties and Metabolic Fate

The norsesquiterpene glycoside ptaquiloside has been extensively investigated for its potential carcinogenic properties [1,102,103,104,105]. Ingesting large quantities or prolonged consumption of ptaquiloside can be toxic to humans, emphasizing the need for caution when considering the edibility of bracken fern. The prevalence of bracken fern in Japanese cuisine is posited as a potential explanation for the increased incidence of bladder, stomach, and esophageal cancers among young Japanese individuals [106,107,108].

Ptaquiloside is implicated as a major contributor to the observed toxicity in ruminant farm animals. Intravenous administration of ptaquiloside to sheep induces progressive retinal degeneration, highlighting its harmful effects [109]. Guinea pigs, unlike rats or mice, exhibit hemorrhagic cystitis and hematuria upon subcutaneous administration of ptaquiloside [110]. It has been suggested that ptaquiloside accounts for more than half of the mutagenic potency associated with bracken [1,102,105]. Although a comprehensive modern carcinogenicity bioassay of ptaquiloside is lacking, limited studies in rats have explored its carcinogenicity. Rats receiving an initial oral dose of 780 mg/kg bw followed by weekly doses of 100 to 200 mg/kg bw or twice-weekly doses of 100 to 150 mg/kg bw for 8½ weeks developed tumors of the mammary gland and ileum, along with other adverse effects [20,111]. In a separate study, rats fed a diet containing 0.027 to 0.08% ptaquiloside developed cancers of the ileum and/or bladder within 15 to 60 days [112]. While parenteral dosing showed no tumors in rats given weekly intravenous doses, adverse effects such as monocytosis and focal renal tubular necrosis were observed [113]. Oral dosing, although not resulting in tumors, led to adverse effects, including tubular necrosis, monocytosis, and elevated plasma tumor necrosis factor-alpha [114].

The mutagenicity of ptaquiloside was tested under different pH conditions. It was found to be nonmutagenic in strains of *Salmonella typhimurium* at pH 7.4 without metabolic activation but exhibited mutagenicity at pH 8.5 after pre-incubation [26,115]. Moreover, ptaquiloside induced chromatid exchange-type aberrations in Chinese hamster lung cells at various pH values, with higher pH values showing greater genotoxic potency [72,116]. In vitro, ptaquiloside produced DNA adducts and induced unscheduled DNA synthesis in a rat hepatocyte culture at pH 7.2 [30,117,118]. Cumulatively, the available evidence suggests that ptaquiloside likely contributes to bracken’s carcinogenicity and toxicity, with a genotoxic mode of action being implicated.

Ptaquiloside undergoes diverse decomposition processes influenced by environmental conditions, particularly pH and temperature [119,120]. Degradation can occur in both acidic and alkaline environments, posing a heightened risk of leaching when soil pH is slightly acidic to neutral, clay content is high, and microbiological degradation is slow [121,122]. Notably, the occurrence of ptaquiloside-induced bovine esophageal and bladder tumors correlates with pH values of 8.1 to 8.2 (saliva) and 7.5 to 8.5 (urine) [71,123].

Under normal physiological conditions, enzymatic degradation of ptaquiloside occurs in the gastrointestinal tract, especially in the stomach and small intestine, catalyzed by glycosidase enzymes. The stomach’s low pH facilitates the hydrolysis of ptaquiloside, leading to the formation of an aglycon called ptaquilosine and a dienone intermediate known as ptaquilodienone (Figure 4) [124,125,126]. Subsequently, two possible paths emerge.

In one scenario, ptaquilodienone reacts with water to produce an aromatic compound, pterosin B, devoid of significant biological effects [72]. These pterosins enter the bloodstream and are transported to the liver, where a series of enzymatic reactions, involving glucosidases, cytochrome P450 enzymes, sulfotransferases, and glucuronosyltransferases, metabolize the breakdown products. This process enhances water solubility, facilitating excretion, typically through urine [102,127,128,129].

Alternatively, ptaquilodienone may covalently bind to DNA molecules [10]. This dienone primarily alkylates amino acids such as cysteine, glutathione, and methionine at their thiol groups. Additionally, minor alkylation occurs at the carboxylate groups, forming corresponding esters [4,72,130]. The dienone also reacts with DNA, targeting adenine (predominantly at *N*-3) and guanine (primarily at *N*-7) residues, resulting in the formation of DNA adducts. This alkylation induces spontaneous depurination and DNA cleavage at adenine base sites [72,131,132]. These alkylated DNA by-products play a crucial role in carcinogenesis, as they introduce mutations and aberrations, leading to the development of cancers. Importantly, if the ptaquiloside molecule degrades to pterosin B, these harmful consequences are averted [102,132,133].

### 4.5. Thiaminase, Caffeic Acid, and Cyanogenic Glycosides: Assessing Bracken Fern’s Metabolic Challenges in Animals

Thiaminases, enzymes with antithiamine properties, are implicated in the short- to medium-term symptoms of bracken fern poisoning in monogastric animals. Unlike ruminants that produce their own thiamine, monogastric animals rely on dietary sources, rendering them more susceptible to thiaminase impact from various origins, including bracken [134]. Thiamine (vitamin B_1_) is sourced by humans from yeast, whole grains (e.g., brown rice, whole wheat), legumes (e.g., lentils, soybeans), nuts and seeds (e.g., sunflower seeds, macadamia nuts), and animal-derived options such as pork and organ meats (liver, kidney, and heart). Fortified foods, particularly cereals and bread, and thiamine-based supplements contribute to thiamine intake in case of dietary insufficiency [135]. Vitamin B_1_ plays a crucial role in maintaining myelin in the peripheral nervous system and overall organism functioning. Its deficiency, leading to beriberi in humans, manifests as peripheral or central neuropathy with symptoms including poor health, weight loss, and loss of coordination, with reported cases suggestive of heart failure [136,137,138].

Monogastric animals acquire vitamin B_1_ through forage diets, with common feed components such as cereal grains (e.g., corn, wheat), oilseed meals (e.g., soybean meal), and specific animal by-products serving as thiamine sources. Animal diets are meticulously formulated to meet thiamine requirements for optimal growth and health [139].

Thiaminase enzymes degrade thiamine into thiazole and pyrimidine. Thiamine deficiency restricts reactions dependent on thiamin diphosphate, leading to substrate accumulation (pyruvate, pentoses, etc.,) associated with these reactions [6,18,46,140]. While thiaminases are heat-stable, autoclaving effectively destroys them [140]. Within *P. aquilinum*, thiaminases contribute to thiamine deficiency, with the highest activity observed in rhizomes and very young fronds, the former having concentrations 10–30 times greater than mature fronds [141,142].

Rats fed on bracken displaying thiaminase activity developed nervous system lesions typical of antivitaminosis B_1_, curable with thiamine therapy [143]. Similar effects were observed in other monogastric animals, such as horses and pigs [19,140]. A 52-week feeding study in rats aimed to explore bracken’s antithiamine activity and its potential link to carcinogenicity. Control groups showed no tumors, while all surviving rats in bracken-fed groups developed multiple intestinal tumors. Bladder tumors occurred in 11% of males and 7% of females in the bracken-alone group, increasing to 53% of males and 67% of females when thiamine was supplemented. Intriguingly, thiamine supplements did not reduce tumor incidence, indicating thiaminase might not be the primary cause of bracken’s carcinogenicity [142,144].

*P. aquilinum* collected in November in the Pretoria district, South Africa, exhibited mean thiaminase activities of 3.1 and 3.5 µg thiamine destroyed/g plant material/hour for types 1 and 2, respectively (Table 1), with a thiamine content of 0.04 µg thiamine/g plant material [18,46,83]. No reports of thiamine deficiency in human populations consuming bracken have been found, potentially attributed to humans having more varied diets and consuming significantly less bracken than animals [134,145]. Another heat-stable antithiamine factor, potentially caffeic acid, has been linked to bracken-induced thiamine deficiency in horses [146,147]. Caffeic acid, a phenolic substance with antioxidant properties found in many plants [148], has shown diverse effects in animal studies, including stomach papillomas in rats and a reduction in colon tumor growth in the same rats [149].

Another potentially hazardous compound occasionally found in specific bracken fern varieties is the cyanogenic glycoside prunasin. While generally present in harmless quantities in bracken fronds, instances of sudden animal death due to suspected hydrogen cyanide (HCN) poisoning from young bracken fronds have been documented [21,147]. Cyanogenesis occurs in both gametophytic and sporophyte generations of ferns, with young fronds being more cyanogenic than older ones [150,151], a typical defense strategy against herbivores [152]. Cyanogenic glycosides, such as prunasin, become toxic through conversion into HCN by *β*-glycosidase enzymes released upon tissue damage. When bracken is crushed or chewed, this conversion occurs, leading to the release of the unstable *α*-cyanohydrin mandelonitrile, which undergoes mandelonitrile *β*-elimination catalyzed via oxynitrilase to produce HCN and benzaldehyde [97,153]. Notably, polymorphism exists in bracken, with some plants lacking either prunasin or the required enzymes, rendering them noncyanogenic [79]. No reports exist of cyanide poisoning in people consuming foods from animals grazing on bracken, indicating animals are more affected by direct prunasin and cyanide exposure.

### 4.6. Ptaquiloside Accumulation and Health Risks: Impact on Animals and the Food Chain

*P. aquilinum* is a widespread plant thriving in diverse environments and posing challenges for pasture control [22]. Documented cases of intoxication after bracken fern ingestion are prevalent in various animals, particularly ruminants such as cattle, sheep, and wild deer [154]. In cattle, ingestion leads to enzootic hematuria, characterized by bleeding into the lower urinary tract and the development of epithelial and mesenchymal neoplasms. Sheep, on the other hand, may experience acute hemorrhagic syndrome, bone marrow aplasia, retinal atrophy, and polioencephalomalacia [155]. While relatively poorly documented in monogastric species, horses, for instance, can suffer from poisoning when exposed to bracken fern through low-quality hay or grazing. This leads to a decrease in thiamine content and an increase in pyruvate and lactate levels in the blood [18,156].

Upon ingestion, ptaquiloside, a key compound in bracken fern, is absorbed into the bloodstream from the stomach and intestines without significant metabolic alterations. Ptaquiloside has the potential to accumulate in various tissues, including the mammary glands, a phenomenon observed during lactation [157]. Since the mammary glands actively produce milk, compounds present in the bloodstream, including ptaquiloside, can contaminate the milk and meat of these animals [128,129,158,159]. The quantity of bracken fern consumed directly correlates with the potential for ptaquiloside accumulation in the animal’s tissues, milk, or meat. Animals engaging in prolonged grazing on *P. aquilinum* are prone to accumulating higher levels of ptaquiloside in their systems. Research indicates that ptaquiloside is excreted in milk at a concentration of 8.6 ± 1.2% of the ingested amount by the cow from the fern, and this excretion is linearly dependent on the dose. Ptaquiloside was detected in milk 38 h after the initial feeding of cows with this plant [159]. Milk from cows fed on bracken has been associated with various adverse effects. These include leukopenia in calves, bladder cancer in mice, and the development of intestinal, bladder, and kidney cancers in rats [20].

Estimates suggest that infants consuming milk from cows exposed to bracken could potentially receive up to 22.8 mg/person/day of ptaquiloside, with elderly individuals and toddlers having the highest per capita chronic milk intake. These estimations, ranging from 2.9 to 22.1 mg/person/day (0.047–0.36 mg/kg bw/day) for toddlers and 2.8 to 21.6 mg/person/day (0.19–1.49 mg/kg bw/day) for the elderly, consider the maximum intake from nontoxic doses of ptaquiloside in milk. However, the actual exposure may be higher when considering milk from poisoned cows or other sources such as drinking water and bracken-exposed animal-derived foods [23,160,161].

Various animal species and individuals metabolize and eliminate ptaquiloside at different rates, influencing susceptibility to its toxicity. Importantly, the consumption of milk or meat from animals exposed to bracken fern poses health risks to humans. Therefore, preventing animal exposure to *P. aquilinum* or its contaminated products is crucial to mitigate the potential transfer of ptaquiloside to the food chain [126,147,162].

## 5. Conclusions

This review critically explored the multifaceted attributes of bracken fern (*Pteridium aquilinum*), delving into its wide-ranging ecological impacts, inherent toxicity, and the intriguing possibilities of its pharmacological attributes. Some bracken fern compounds demonstrate antimicrobial, antibacterial, and antioxidant properties, hinting at their therapeutic potential. However, it is imperative to highlight that *P*. *aquilinum* is also associated with the presence of ptaquiloside, a compound with potentially carcinogenic properties. It poses a threat to both animals exposed to the plant and humans, who may intake this compound through contaminated foods, particularly milk and meat. Beyond the well-studied ptaquiloside, attention should also be directed toward other compounds contributing to the overall toxicity of the plant, such as thiaminase enzymes and prunasin. Thiaminases, known for their antithiamine properties, introduce short- to medium-term health challenges in monogastric animals. While they pose a risk, especially in the rhizomes and young fronds, exploration of the varied impacts and potential mitigation strategies is essential. Similarly, prunasin, a cyanogenic glycoside, adds another layer of consideration, with the release of hydrogen cyanide upon plant tissue damage. The preparation and cooking methods applied to bracken fern have a discernible impact on the concentration of these compounds. Various studies indicate that heat treatment or specific cooking techniques, such as boiling or steaming, may reduce or eliminate ptaquiloside and thiaminases. Overall, *P*. *aquilinum*, as a medicinal plant, is characterized by not only positive but also negative aspects. Understanding the ecological implications of its proliferation, devising innovative control strategies, and exploring its pharmaceutical potential offer exciting prospects. Moreover, a deeper understanding of ptaquiloside’s metabolism and the development of methods to reduce its presence in food and water sources could revolutionize food safety practices. In essence, this review aimed not only to provide a comprehensive synthesis of existing knowledge but also to inspire future inquiries. To enhance the effectiveness of controlling bracken fern’s invasive spread and mitigating its negative effects, it is crucial to focus on developing sustainable management practices. This could include targeted removal of bracken fern in sensitive ecological areas, promoting the growth of native vegetation, and employing biological control methods. Furthermore, public awareness campaigns about the potential health risks associated with *P*. *aquilinum* consumption, especially in livestock and derived products, are essential to mitigate human exposure to harmful compounds such as ptaquiloside. Critically evaluating the strengths and weaknesses of the previously submitted literature review text emphasizes the need for continuous research and updates in this dynamic field. Regular reassessment of the ecological impact, pharmacological potential, and toxicity of bracken fern will contribute to a more nuanced understanding and to inform management strategies.

## Figures and Tables

**Figure 1 plants-13-00018-f001:**
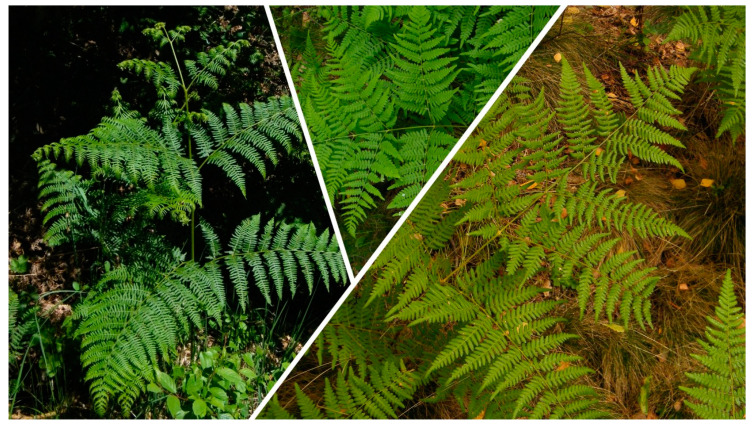
Bracken fern growing in the wild. Photos by Zdeňka Navrátilová.

**Figure 2 plants-13-00018-f002:**
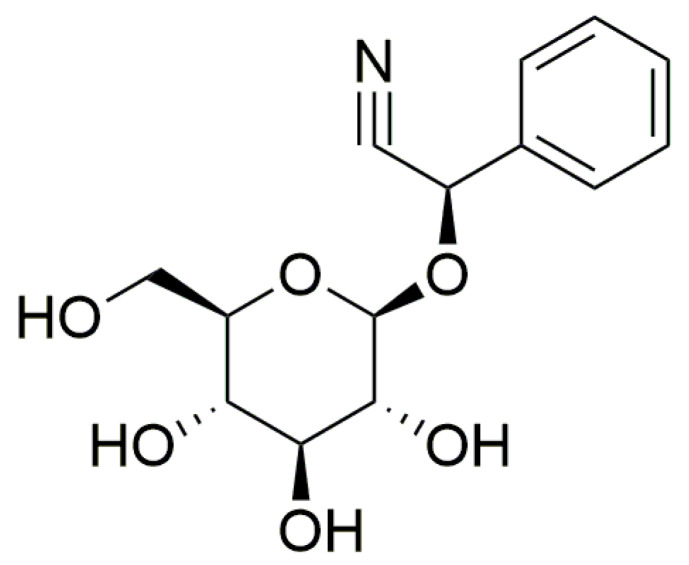
The chemical structure of the cyanogenic compound prunasin.

**Figure 3 plants-13-00018-f003:**
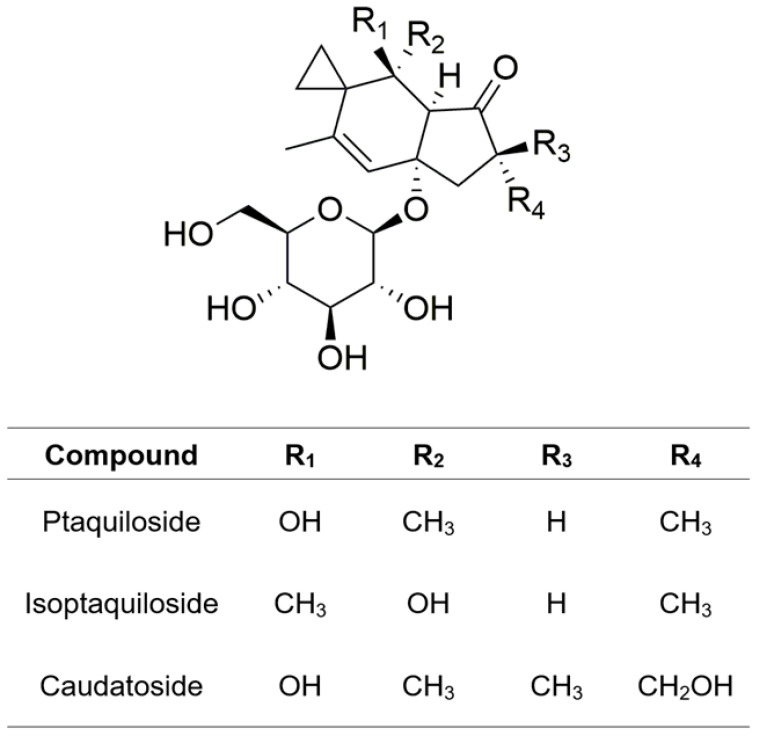
Chemical structures of illudane glycosides.

**Figure 4 plants-13-00018-f004:**
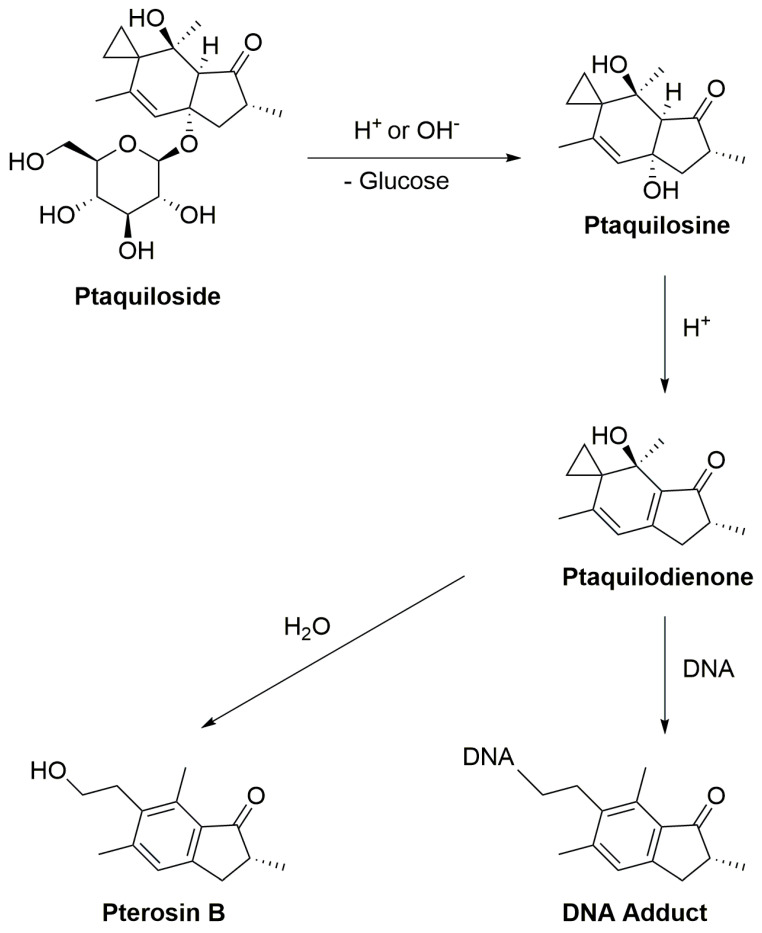
The scheme of ptaquiloside degradation reactions.

## Data Availability

Data are contained within the article.

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
