# Peer review of "Health and Environmental Hazards of the Toxic Pteridium aquilinum (L.) Kuhn (Bracken Fern)"

_plants, 2023, doi:10.3390/plants13010018_

Round 1

Reviewer 1 Report

Comments and Suggestions for Authors

A very comprehensive balanced review of bracken fern, a ubiquitous global presence. Besides its ecological competitiveness and impacts on other plants, various phytochemicals by this plant are well reviewed. While some hold good promise for pharmacological effects, others pose challenges to animal and human health through food contamination. Ptaquiloside and thiaminase and their impact on animal and human health are thoroughly reviewed and discussed. The manuscript is well written with good flow.

In view of the conclusion that this fern does more harm than good, the reviewer suggests that the authors discuss ways to mitigate bracken fern or control its invasive spreading.

Reviewer 2 Report

Comments and Suggestions for Authors     The review manuscript addresses different important aspects of the species, including morphological, toxicological, pharmacological, and chemical composition aspects. The methodology used in the review is well defined. The species is widely distributed worldwide, which makes it of interest to different communities. Tables and figures presented adequately illustrate the review. I suggest that the illustrative figure be inserted in the introduction.

Reviewer 3 Report

Comments and Suggestions for Authors

Overall, Malík et al., is interesting and summarized literature data for Pteridium aquilinum. The article is well-written and after revision it will be good for publications. I have more technical suggestions.

Title – in my opinion it is good if authors considered to change the popular name of plant with Latin name. For example, “Health and Environmental Hazard of the Toxic Pteridium aquilinum (L.) Kuhn(Bracken Fern).

Abstract

Please rewrite it and add information (1) period of research and databases used; (2) the main problem after literature report; (3) the main results; (4) conclusion;

Introduction

Overall, this section should be split into subsection so it will be easier to read OR move information from this section to subsequent sections. Furthermore, the authors should to clarify what is the issues with the species; what are the general problems; ect. According to Title of article, the main goal in this article is to health and environmental hazard…but not taxonomical.

L35 – please cut the plan name.

L35-50 is the Botanical or taxonomical or ecological features of species. But section 2. the authors were wrote: Botanical aspects of the plant. It is confusing. Please clarify and so it will be easier to read.

L102 – please clarify how long is different periods.

Figure 1 – please add information: (1) are these your pictures or did you get them from somewhere?

Figures 2; 3; 4 – please specify the origin of the chemical structure is yours or …what?

Conclusion

It is good if authors rewrite this section and critically evaluate the strengths and weaknesses of the literature review. In my opinion, it is not good to repeat data that has already been written above.

Round 2

Reviewer 3 Report

Comments and Suggestions for Authors

The authors have followed my recommendations. I have no comments.